# Effects of Dietary Supplementation of gEGF on the Growth Performance and Immunity of Broilers

**DOI:** 10.3390/ani11051394

**Published:** 2021-05-13

**Authors:** Jianyong Zhou, Jingyi Yao, Luhong Bai, Chuansong Sun, Jianjun Lu

**Affiliations:** 1Hainan Institute of Zhejiang University, Yongyou Industry Park, Yazhou Bay Sci-Tech City, Sanya 572000, China; 22017073@zju.edu.cn; 2Key Laboratory of Molecular Animal Nutrition (Zhejiang University), Ministry of Education, Key Laboratory of Animal Feed and Nutrition of Zhejiang Province, Institute of Feed Science, College of Animal Science, Zhejiang University, Hangzhou 310058, China; jyy21@zju.edu.cn (J.Y.); 21817070@zju.edu.cn (L.B.); 13864091954@163.com (C.S.)

**Keywords:** gEGF, antibiotic replacement, broilers, growth performance, antioxidant capacity, immunity

## Abstract

**Simple Summary:**

Epidermal growth factor (EGF), a milk-borne growth factor, has been proved to stimulate the growth of animals. However, the current yield of epidermal growth factor from various sources remain low. The goal of this study was to determine at what age chicken embryos have the highest EGF content and then to extract that EGF and include it in the diet of broiler chickens in order to assess differences in growth, feed efficiency, serum metabolites, antioxidant capacity and immune status. Overall, our results showed that including EGF harvested from chicken embryos in the diet of broilers had beneficial effects on production.

**Abstract:**

EGF has been shown to stimulate the growth of animals. In this study, the content of EGF in chicken embryos (gallus EGF, gEGF) aged from 1 to 20 days of incubation were determined by ELISA kit, and the 5-day-old chicken embryos with the highest content of 5593 pg/g were selected to make gEGF crude extracts. A total of 1500 1-day-old Xianju chickens were randomly divided into five groups with six replicates of 50 chickens each. The control group was fed a basal diet, and other treatment diets were supplemented with 4, 8, 16 and 32 ng/kg gEGF crude extract, respectively. The experiment lasted for 30 days. Chicks were harvested at the end of the experiment, and liver, spleen, thymus, bursa and serum samples were collected. Results showed that average daily gain (ADG) and average daily feed intake (ADFI) of 16 ng/kg group were higher than those in the control group (*p* < 0.05). The serum uric acid (UA) of the 16 ng/kg group was reduced (*p* < 0.01), and the serum alkaline phosphatase (AKP) of the 16 ng/kg group increased (*p* < 0.01). The gEGF extract also increased chick’s antioxidant capacity, decreased malondialdehyde (MDA) and increased catalase (CAT) in the liver and serum of 16 ng/kg groups in compared to the control group (*p* < 0.01). Furthermore, immunity was improved by the addition of gEGF to broiler diets. The serum immunoglobin A (IgA) content of 8 and 16 ng/kg groups and the serum immunoglobin M (IgM) content of 4 and 8 ng/kg groups were increased (*p* < 0.05) compared to the control group. The bursa index of each experimental group was higher than the control group (*p* < 0.01). These findings demonstrate that the crude extract of gEGF prepared in this experiment could improve the growth performance, antioxidant capacity and immunity of broilers.

## 1. Introduction

Chicken meat is recognized for its several health benefits due to its high nutritional value and relatively lower content of fat and cholesterol compared with red meat, such as pork, beef and mutton [1]. Combined with the lower price and rare religious restrictions, the consumption demand of chicken is increasing gradually [2]. In recent years, the poultry industry has been developing rapidly, spurred by increased consumption of poultry meat and by-products [3]. Antibiotics are routinely used to maintain the health and productivity of chickens and pigs in many low- and middle-income countries, due to the massive expansion of intensive animal production systems [4]. However, the abuse of antibiotics in animals can also decrease immune function, causing a decrease in disease resistance [5,6]. It was found that, when broilers were fed with antibiotics for 42 days, the number and diversity of beneficial bacteria in their intestines decreased, the intestinal microorganisms did not mature until the 40th day and the content of immunoglobulin in serum of broilers decreased [7].

According to the General Assembly of the United Nations, the phenomenon of antibiotic resistance is a priority topic for human development, being on par with global warming [8]. Although most antibiotic resistance in humans comes from antibiotics used by humans, in the context of the ban on the use of antibiotics in livestock production, a number of studies have focused on the development of alternatives to antibiotics to maintain the production performance and health of farmed animals [9,10].

Many milk-borne growth factors, such as EGF, insulin, glucagon-like peptide-2 and insulin-like growth factor, have been shown to improve the morphology and function of intestine [11,12,13]. Maintaining gut health is essential for improving feed efficiency, promoting growth performance and ensuring the overall health of animals [14,15]. EGF is one of the most abundant milk-borne growth factors, which plays an important role in animal gut development and is considered to be the main nutritional factor regulating intestinal development and maturation. Several studies have found that EGF can improve the quality and length of small intestine and colon, promote the differentiation of goblet cells, endocrine cells and Paneth cells, and increase the digestibility of nutrients and the mRNA expression of nutrient transporters [16,17,18,19,20].

Initially, researchers isolated and extracted EGF from human urine and shrew submandibular glands, and later isolated EGF from human and other animals’ breast milk, saliva, tears and plasma [21,22,23,24]. However, current yields remain low and novel extraction pathways need to be explored. With the increasing development of genetic engineering technology, people began to use exogenous expression systems to express EGF. Lee et al. [25] successfully expressed recombinant porcine EGF (pEGF) in Pichia pastoris expression system and proved its promoting effect on the development of the digestive tract in weaned piglets. In addition, some researchers successfully expressed recombinant EGF in *Escherichia coli* and *Lactococcus lactis* [12,26]. However, its treatment process is complex, the required culture environment is harsh and it is still at an early stage.

At present, the research on EGF is mainly focused on human, pig, mouse and other mammals; there are few reports on poultry EGF. As such, we hypothesized that EGF could be extracted from chicken embryos, and it may improve the growth and performance of broilers when fed orally. The present study had two main goals. First, we wanted to assess at what age the EGF content was the highest in chicken embryos. Second, we determined the effects of including the extracted gEGF in the diet of broilers on growth, feed efficiency, serum metabolites and immune status.

## 2. Materials and Method

All procedures of animal experiments were carried out based on protocols approved by the Animal Care Advisory Committee of Zhejiang University (No. ZJU2013105002, Hangzhou, China). The birds used in the current study were treated humanely. Great efforts were made to minimize suffering.

### 2.1. EGF Content of Chicken Embryos during the Incubation

The experimental procedures were conducted with reference to the methods in a previously published patent [27]. A total of 120 embryos of Xianju chickens aged from 1 to 20 days of incubation were tested, six of each age. Chicken embryos were shelled, and 0.05 mol/L acetic acid (4 °C) was added to homogenate the tissues. The prepared homogenate was centrifuged at 2500× *g*, at 4 °C, for 20–30 min, and the supernatant was collected. The precipitation was further homogenized with 0.05 mol/L acetic acid (4 °C) and then centrifuged at 2500× *g*, for 20–30 min, at 4 °C. The supernatants of the two homogenates were mixed, and fat was removed by centrifugation. Then, the mixed supernatant was slowly adjusted to pH 7.0 by adding (NH4)_2_SO_4_. EGF concentration in the supernatant was determined by using a commercially available ELISA kit (Shanghai Enzymatic Biotechnology Co., Ltd., Shanghai, China).

### 2.2. Crude Isolation of gEGF

According to the EGF content of chicken embryos on different days obtained by ELISA kit in Section 2.1, the isolation experiments were carried out on the chicken embryos of 5-day-old with the highest EGF concentration. The sodium benzoate (25 g/L) was added to the final supernatant obtained in Section 2.1 and stirred until completely dissolved. The solution was adjusted pH to 4.6 by adding acetic acid (4 °C), then stirred for half an hour and filtered under reduced pressure. The precipitates were then dried at room temperature, and 20 mL acetone (4 °C) was added per gram of precipitation. After standing overnight, decompression and filtration were performed. Repeating the above steps several times, the crude extract of powdered gEGF with a purity of about 80% was obtained. The purity is calculated by the percentage of EGF content in the total organic matter. The total carbon content of the accumulated organic matter was determined by using total organic carbon (TOC) analyzer (TOC-VCPH, Shimadzu, Japan).

### 2.3. Effect of Crude Extract of gEGF on the Growth and Development of Broilers

#### 2.3.1. Animals and Experimental Treatments

The experimental animals were 1-day-old Xianju chickens, provided by Ningbo Zhenning Animal Husbandry Co., Ltd (Ningbo, Zhejiang, China). The test site was Animal Husbandry breeding Center in Ningbo, Zhejiang Province. A total of 1500 1-day-old healthy Xianju chickens with similar weight were randomly divided into 5 groups, one of which was randomly selected as the control group, and the other 4 groups were selected as the treatment group. There were 6 replicates in each group, with 50 birds each. The control group was fed with the basic diet produced by the feed factory of Ningbo Animal Husbandry Co., Ltd., while the treatment group was fed with the basic diet supplemented with 4, 8, 16 and 32 ng/kg gEGF crude extract (the actual content of gEGF, not the total amount of crude extract). The composition and nutrition level of the experimental basic diet were shown in Table 1. The experiment lasted for 30 days. Each group was reared in different cages (120 × 50 × 45 cm^3^, length × width × height) in the same breeding house to ensure the same feeding environment. All chicks were fed 6–8 times per day in the first week and 3–4 times per day after 2 weeks of age. Continuous light was provided 24 h for first 3 days, and the temperature was maintained at 34 ± 1 °C. After that, the temperature was then gradually decreased per week, and the light decreased to 12 h per day. The humidity was maintained at 70–75% for the first week, at 65% for the second week and 50–60% for the remainder of the study. Free feeding and drinking water during the whole period, preventive immunization and feeding management were carried out in accordance with conventional methods.

#### 2.3.2. Body Measurements and Blood Sampling

During the experiment, with repetition as the unit, feed intake of each repetition was recorded every day, and ADFI was calculated. ADG was calculated based on the difference between the end and the beginning weights of the experimental period. Calculation of the feed-to-gain ratio (F/g) was based on ADG and ADFI. After regular feeding, all the chickens were fasted for 12 h before slaughter, with free access to water. Then 2 chicks were randomly selected from each repeat (12 chicks in each group; a total of 60 chicks) for slaughtering and sampling. Blood was collected from jugular vein and placed in the procoagulant tube. After standing for 30 min, blood samples were centrifuged for 10 min at 3000× *g* to separate the serum, and then they were divided into Eppendorf tubes of 1.5 mL and stored in a freezer, at −80 °C, for use.

#### 2.3.3. Serum Biochemical Indices

Serum samples were incubated on ice for 10 min and centrifuged for 10 min at 3000× *g*. The contents or activities of total protein (TP), albumin (ALB), uric acid (UA), blood urea nitrogen (BUN), calcium (Ca), phosphorus (P), alkaline phosphatase (AKP), glutamic pyruvic transaminase (GPT) and glutamic oxaloacetic transaminase (GOT) in serum were determined by kits, which were purchased from Nanjing Jiancheng Institute of Biological Engineering. The measuring operation was carried out according to the instructions. A microplate reader (Biotek ELX800; Biotek Instruments, Inc., Winooski, VT, USA) was used in the determination.

#### 2.3.4. Antioxidant Capacity

Liver tissue samples were cut into small pieces (about 0.1 g), and then ice-cold physiological saline was added to it at a ratio of 1:9 to prepare 10% tissue homogenate. The homogenate was centrifuged at 3000× *g*, for 10 min, at 4 °C. The supernatant was collected and stored at −80 °C for the following analysis. The contents or activities of total antioxidant capacity (T-AOC), total superoxide dismutase (T-SOD), catalase (CAT) and malondialdehyde (MDA) in the serum and hepatic supernatants were assayed by using commercially available assay kits (Nanjing Jiancheng Institute of Biological Engineering, Nanjing, China) as per the instructions of the manufacturer. The optical density (OD) value of each sample was measured by spectrophotometer (UV-1601 UV–VIS Spectrophotometer, Shimadzu Corporation, Tokyo, Japan).

#### 2.3.5. Immune Performance

The spleen, thymus and bursa were removed and weighed after the experimental chickens were slaughtered. Calculate the immune organ index (g/kg) according to the formula of (immune organ weight)/(live weights before slaughter).

The total IgG, IgA and IgM in serum were measured with commercially available assay kits (Shanghai ELISA Biological Technology Co., Ltd., Shanghai, China) as per the instructions of the manufacturer. A microplate reader (Biotek ELX800; Biotek Instruments, Inc., Winooski, VT, USA) was used in the determination.

### 2.4. Statistical Analysis

The experimental data were analyzed by one-way ANOVA with SPSS 21.0 software (SPSS Inc., Chicago, IL, USA). Cage served as the experimental unit (*n* = 6). Pairwise comparisons were investigated with Fisher’s LSD test and subsequent Holm-Bonferroni correction for multiple comparisons. Mean values were further examined with Duncan’s multiple comparisons to determine statistical differences among treatment groups. A statistical significance was declared at *p* < 0.05 or *p* < 0.01.

## 3. Results

### 3.1. EGF Content of Chicken Embryos during the Incubation

As presented in Figure 1, there were significant changes observed in the content of EGF in chicken embryos of different days. The content of gEGF increased significantly from 1- to 5-day-old chicken embryos reaching the highest value of 5593 pg/g at 5 days old, whereas it decreased significantly at 6 days old and remained at relatively stable levels afterward.

### 3.2. Growth Performance

Compared with the control group, dietary supplementation with 4 and 16 ng/kg gEGF crude extract groups increased the ADG and the ADFI (*p* < 0.05; Table 2). There were no significant differences in F/g between the group supplemented with gEGF and control group (*p* > 0.05).

### 3.3. Serum Biochemical Indices

The serum UA concentration of each gEGF group was lower compared with the control group (*p* < 0.01), and the serum AKP concentration of 8 and 16 ng/kg gEGF groups increased significantly (*p* < 0.01). The serum GOT concentration of 4, 8 and 16 ng/kg groups was lower than that in the control group (*p* < 0.01). There were no significant differences in serum TP, ALB, BUN, Ca, P and GPT levels among all treatment groups (*p* > 0.05; Table 3).

### 3.4. Antioxidant Capacity Indices

As shown in Figure 2 and Figure 3, compared with the control group, there was no significant difference in T-AOC level and serum T-SOD activity (*p* > 0.05). In the 4 ng/kg group, the activity of T-SOD and CAT in the liver was increased (*p* < 0.01), and the level of MDA in serum and liver was decreased (*p* < 0.01). Dietary supplementation of 16 ng/kg gEGF crude extract could increase the activity of CAT in serum and liver, and decrease the level of MDA in serum and liver (*p* < 0.01).

### 3.5. Immune Performance

The serum IgA content of 8 and 16 ng/kg and the serum IgM content of 4 and 8 ng/kg groups were increased (*p* < 0.05) compared to the control group (Table 4). There was no significant difference in the content of serum IgG between treatments and control (*p* > 0.05).

The results showed that the bursa index of each experimental group was higher than the control group (*p* < 0.01; Figure 4). Except for 32 ng/kg group, the other experimental groups had no significant effect on thymus index and spleen index compared with the control group (*p* > 0.05).

## 4. Discussion

The change of gEGF content may be related to the development of various organs in the chicken embryo. On day 1 of incubation, a three-germ structure begins to form in the chicken embryo. On day 2 of incubation, the original heart is formed, and the endoderm forms the foregut [28], a pocket-like cavity, which will develop into the front part of the digestive tract (oral cavity, esophagus, proventriculus and muscular stomach) and liver and pancreas. When chicken embryo hatches on day 7 or 8 of incubation, the structure and function of the intestine, pancreas and liver are basically perfect. From the perspective of chicken embryo development, the digestive system is one of the earliest organs formed by the chicken embryo [29,30,31]. Previous studies have found that the concentration of EGF in the digestive tract is much higher than that in the circulation. During the piglet weaning process, the dietary transition from breast milk to solid feed will lead to the loss of EGF, which will result in the shortening of the piglet’s villus height and reduced digestibility [32]. Duh et al. [33] used EGF to culture early mouse embryos and found that intestinal columnar epithelium, goblet cells and intestinal villi formed earlier in the experimental group. Therefore, it can be inferred that EGF is closely related to the development of digestive tract. This may be the reason for the significantly increased content of gEGF shown in Figure 1 during 1–5 days of incubation. After the 6th day, due to the simultaneous development and basic formation of multiple digestive organs, the content of gEGF decreased sharply, which may only play a role in promoting the formation of surface epithelial cells and regulating cell differentiation.

Some researchers [34,35] added EGF expressed by *Saccharomyces cerevisiae* to the diet of weaned piglets and found that the ADG of the experimental group was significantly higher than that of the control group. In this study, ADG and ADFI of broilers significantly increased in the groups of 4 and 16 ng/kg gEGF crude extract compared with the control group, agreeing with previous findings. Bedford [12] studied the effect of pEGF on the growth performance of pigs and found that pEGF could not significantly reduce the F/g after 2 weeks. There was also no significant improvement of F/g in our study.

Blood biochemical indices can reflect the changes of metabolic, growth and development status of livestock [36,37]. In this study, there were no significant differences in the contents of TP and ALB among the groups. These results indicated that there was no improvement in the ability of protein synthesis in the liver of broilers under the treatment. UA and BUN are important indexes to evaluate renal function and could reflect the ability of amino acid metabolism [38,39]. The serum UA of each experimental group was significantly lower than that of the control group, indicating that the crude extract of gEGF in diet could improve the utilization of amino acids in broilers to some extent. Dietary supplementation of gEGF crude extract had no significant effect on serum Ca and P levels of broilers compared with the control group, but serum AKP level significantly increased in the 8 and 16 ng/kg gEGF groups, and there was a significant difference between them. AKP can be expressed in liver, bone, intestinal tract and other parts, and it is involved in maintaining the balance of calcium and phosphorus in the body, which is of great significance to the osteogenesis and hepatobiliary function of the body [40,41]. The serum GOT of 4, 8 and 16 ng/kg groups was significantly lower than that of the control group, indicating that the appropriate added dose could improve the liver function of broilers, which was consistent with the previous results of increased serum AKP. Generally speaking, the addition of gEGF crude extract can improve the liver function of broilers and promote the utilization of amino acids.

Antioxidant systems mainly include enzyme antioxidant systems, such as T-AOC, T-SOD, CAT, etc., and non-enzymatic antioxidant systems, such as vitamin E, glutathione and so on [42,43]. Antioxidant capacity is usually evaluated by determining the activity of antioxidant enzymes [44,45]. Tang et al. [46] studied the effects of EGF on the oxidative stress of porcine intestinal epithelial cell line IPEC-J2 induced by lipopolysaccharide. It was found that EGF could significantly reduce the level of MDA in cells and culture medium, and significantly increase the levels of T-AOC, CAT and SOD, thus reducing oxidative damage. The results were similar to those of this experiment, while the differences may be caused by different in vivo and in vitro tests, as well as experimental species and doses. All in all, the crude extract of gEGF could improve the antioxidant capacity of broilers.

Enhancing poultry immune is of great significance for improving poultry health. Poultry immunoglobulins mainly include IgA, IgG and IgM, all of which are produced by B cells [47,48]. Except for the IgG content of the experimental group, IgA and IgM both increased significantly. IgA is recognized as the important antibody isotype involved in protective responses on mucosal surfaces [49]. IgM is the first immune antibody, which can dissolve pathogenic bacteria by activating and controlling the complement classical pathway and inflammation [50]. Lee et al. found that dietary pEGF supplementation could enhance serum IgA in early weaned piglets [25]. Moreover, some studies showed that the serum IgA, IgG and IgM levels of weaned piglets fed with recombinant pEGF expressed by *Saccharomyces cerevisiae* increased significantly [34,51]. These findings parallel those of the present study. Thymus index and spleen index could reflect the capacity of the immune system [52], as well as the bursa of Fabricius, which is a lymphoid organ unique to chickens and it is critical to normal avian B-lineage development [53]. The bursa index of each gEGF group increased significantly, and the thymus index and spleen index increased significantly in 32 ng/kg group. It is also suggested that gEGF crude extract can increase the immunity of broilers by promoting the development of immune organs and increasing the secretion of IgA and IgM.

## 5. Conclusions

This study provided a novel way to obtain EGF. We found that the content of gEGF increased markedly from 1- to 5-day-old chicken embryos, and it reached the peak at 5 days old. The gEGF extracted from 5-day-old chicken embryos was added to the diet of broilers, which proved that gEGF could improve the growth performance, antioxidant capacity and immunity of broilers. Considering all the indexes, the optimum addition amount of gEGF crude extract in the diet of broilers was 16 ng/kg. Chicken embryos can develop into individuals in a short period of 21 days, which shows the strong effect of related growth factors. Moreover, sourcing EGF from chicken embryos has many advantages; they are widely available and inexpensive, which has a great advantage for future mass production of EGF. Nevertheless, further studies on the increase in the gEGF purity of the isolates are needed to be explored.

## Figures and Tables

**Figure 1 animals-11-01394-f001:**
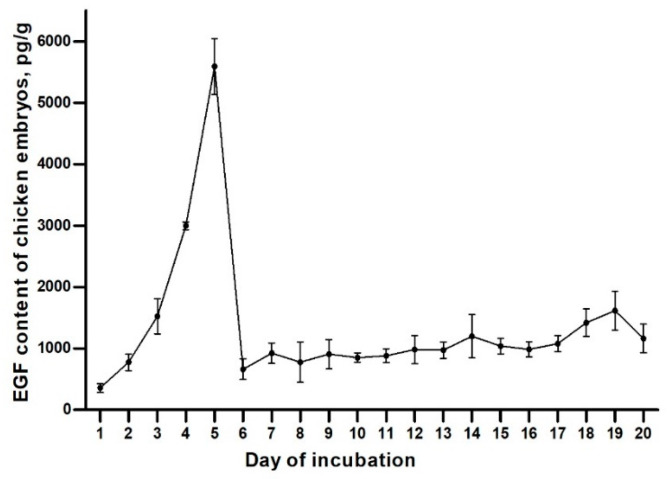
The EGF content of chicken embryos during the incubation. Values are presented as mean, and the bars represent SEM (*n* = 6).

**Figure 2 animals-11-01394-f002:**
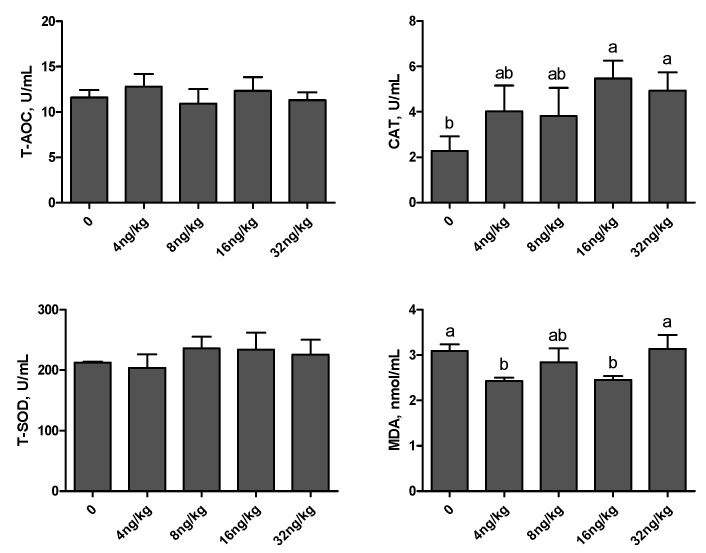
Effects of gEGF crude extracts on antioxidant indexes in the serum of broilers. Column data are presented as mean, and the bars represent SEM (*n* = 6). (**a**,**b**) Means within the same row without common superscripts significantly different (*p* < 0.05). Abbreviations: T-AOC, total antioxidant capacity; T-SOD, total superoxide dismutase; CAT, catalase; MDA, malondialdehyde.

**Figure 3 animals-11-01394-f003:**
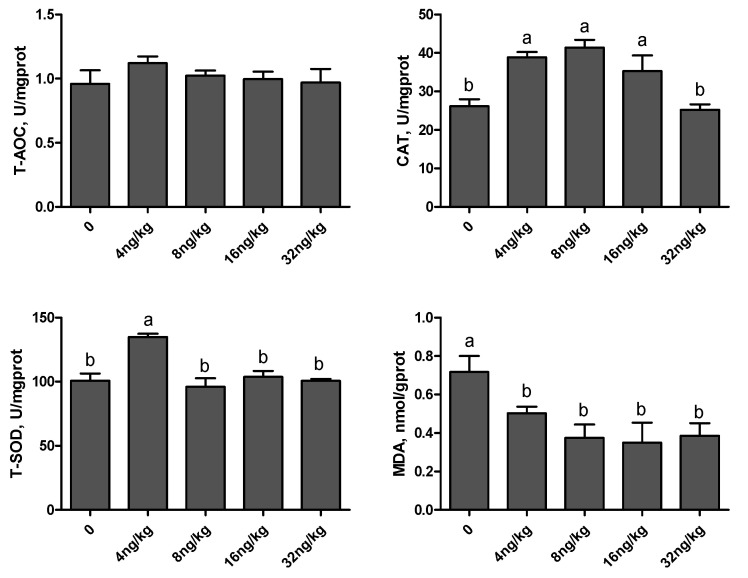
Effects of gEGF crude extracts on antioxidant indexes in liver of broilers. Column data are presented as mean, and the bars represent SEM (*n* = 6). (**a**,**b**) Means within the same row without common superscripts significantly different (*p* < 0.05). Abbreviations: T-AOC, total antioxidant capacity; T-SOD, total superoxide dismutase; CAT, catalase; MDA, malondialdehyde.

**Figure 4 animals-11-01394-f004:**
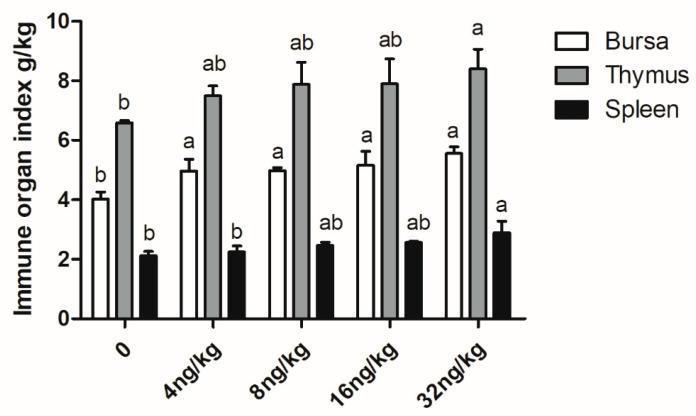
Effects of gEGF crude extracts from chicken embryos on immune organ indexes of broilers. Column data are presented as mean, and the bars represent SEM (*n* = 6). (**a**,**b**) Means within the same row without common superscripts significantly different (*p* < 0.05).

**Table 1 animals-11-01394-t001:** Ingredient compositions and nutrient levels of basal diet for broilers (DM basis).

Ingredients	Content (%)	Nutrition Level ^2^	Content
Corn	62.35	Metabolizable energy, MJ/kg	11.68
Soybean meal	24.00	Crude protein, %	18.90
Wheat middlings	4.00	Lysine, %	0.93
Cottonseed meal	4.00	Methionine, %	0.43
Fish meal	2.00	Threonine, %	0.71
Premix ^1^	1.00	Arginine, %	1.25
CaCO3	1.20	Total phosphorus, %	0.65
CaHPO4	1.10	Calcium, %	0.74
NaCl	0.35		
Total	100.00		

^1^ The premix provided following per kilogram of diet: vitamin A, 1000 IU; vitamin D_3_, 300 IU; vitamin E, 16.5 mg; vitamin K, 2 mg; vitamin B_1_, 1 mg; vitamin B_2_, 8.5 mg; vitamin B_12_, 0.02 mg; folic acid, 0.8 mg; niacin, 50 mg; pantothenic acid, 14 mg; Mn, 60 mg; Zn, 40 mg; Fe, 80 mg; Cu, 8 mg; I, 0.35 mg; Se, 0.15 mg. ^2^ The metabolic energy was a calculated value; the others were measured values.

**Table 2 animals-11-01394-t002:** Effects of gEGF crude extracts on growth performance of broilers from 1 to 30 days of age.

Item	Control	gEGF Crude Extract (ng/kg)				*p*-Value	SEM
		4	8	16	32		
ADG, g	7.44 ^b^	7.83 ^a^	7.65 ^a,b^	7.78 ^a^	7.53 ^a,b^	0.012	0.10
ADFI, g	16.4 ^b^	17.4 ^a^	17.1 ^a,b^	17.3 ^a^	16.7 ^a,b^	0.019	0.29
F/g	2.20	2.22	2.23	2.23	2.21	0.96	0.05

Values represent mean, *n* = 6. ^a,b^ Means within the same row without common superscripts significantly different (*p* < 0.05). Abbreviations: ADG, average daily gain; ADFI, average daily feed intake; F/g, feed-to-gain ratio.

**Table 3 animals-11-01394-t003:** Effects of gEGF crude extracts on serum biochemistry of broilers.

Item	Control	gEGF Crude Extract (ng/kg)				*p*-Value	SEM
		4	8	16	32		
TP, g/L	37.4	38.2	36.3	39.6	37.6	0.39	1.57
ALB, g/L	17.4	18.7	17.6	19.3	17.4	0.13	0.80
UA, mmol/L	1.36 ^a^	1.11 ^b^	1.14 ^b^	0.96 ^b^	1.14 ^b^	0.001	10.67
BUN, mmol/L	2.24	1.82	1.90	1.79	1.90	0.11	0.16
Ca, mmol/L	2.07	2.38	2.25	2.2	2.36	0.15	0.12
P, mmol/L	0.49	0.56	0.51	0.51	0.54	0.23	0.03
AKP, U/L	0.87 ^c^	1.38 ^b,c^	1.72 ^b^	2.76 ^a^	1.52 ^b,c^	<0.001	2.48
GOT, U/L	36.7 ^a^	21.9 ^b^	20.3 ^b^	21.6 ^b^	34.1 ^a^	<0.001	2.39
GPT, U/L	2.21	1.93	1.93	1.90	1.88	0.48	0.29

Values represent mean, *n* = 6. ^a,b,c^ Means within the same row without common superscripts significantly different (*p* < 0.05). Abbreviations: TP, total protein; ALB, albumin; UA, uric acid; BUN, blood urea nitrogen; Ca, calcium; P, phosphorus; AKP, alkaline phosphatase; GOT, glutamic oxaloacetic transaminase; GPT, glutamic pyruvic transaminase.

**Table 4 animals-11-01394-t004:** Effects of gEGF crude extracts on serum immunological indexes of broilers.

Item	Control	gEGF Crude Extract (ng/kg)				*p*-Value	SEM
		4	8	16	32		
IgA, μg/mL	244 ^b^	314 ^a,b^	318 ^a^	342 ^a^	311 ^a,b^	0.011	21.26
IgG, μg/mL	677	711	765	752	738	0.66	63.26
IgM, μg/mL	1788 ^b^	2118 ^a^	2122 ^a^	2104 ^a,b^	2034 ^a,b^	0.028	96.81

Values represent mean, *n* = 6. ^a,b^ Means within the same row without common superscripts significantly different (*p* < 0.05). Abbreviations: IgA, immunoglobin A; IgM, immunoglobin M; immunoglobin G.

## Data Availability

The data presented in this study are available on request from the corresponding author. The data are not publicly available due to privacy.

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
