# Peer review of "Effects of Dietary Supplementation of gEGF on the Growth Performance and Immunity of Broilers"

_animals, 2021, doi:10.3390/ani11051394_

Round 1

Reviewer 1 Report

The manuscript quality has been considerbaly improved. Authors addressed all reviewer's comments and suggestions and provided satisfactory explanations. I have no further comments. The manuscript can be accepted for publication.

Author Response

We thank the reviewer for his/her enthusiasm for our paper, and for recognizing the amount of work involved!

Reviewer 2 Report

Overall, the simple summary is much improved. The last sentence of the simple summary needs to be re-written for clarification. It should read: Overall, our results showed that including EGF harvested from chicken embryos in the diet of broilers had beneficial effects on production. As written it is a little vague.

The abstract is overall improved but needs some further detail. At line 19, after ages, include the ages that were tested and then later in the sentence indicate which age had the highest content. This information is currently in line 25. Throughout the abstract, remove the word significant and simply include the p value. In the sentence beginning in line 31, you need to justify this statement – was there a difference in serum IgA and IgM or are you extrapolating from your other findings? If there was no difference, just state what your results mean. It would be best if you present your results in changes in markers related to antioxidant capacity and/or immunity – this would help your audience follow a little easier.

The authors have done a good job of addressing concerns in the introduction. There are a few specific grammar changes that can be noted below. In addition, in the first sentence of the introduction, the authors need to state what the chicken meat is improved in relative to – beef meat? Pork meat? Other meat? The same comment applied in the sentence beginning on line 42 – lower price relative to what, other meats? In the sentence beginning on line 51, this is only when proper withdrawal times for antibiotics aren’t followed. In most developed countries, meat is checked for antibiotic residue. Most of the antibiotic resistance that occurs in animal-human transmission occurs in the humans that are touching and handling the animals or their waste products. The sentence beginning on Line 81 should read: As such, the authors hypothesize that EGF could be extracted from chicken embryos and it may improve the growth and performance of broilers when fed orally. The sentence(s) beginning on line 86 should read: The present study had two main goals. First, we wanted to assess at what age the EGF content was highest in chicken embryos. Second, we determined the effects of including the extracted gEGF in the diet of broilers on growth, feed efficiency, serum metabolites, and immune status.

Again the materials and methods are improved, but there are still several concerns. First, the grammar needs to be improved throughout – I suggest having this edited by someone who is a native English speaker. At line 190, this reviewer suggest including n=6 after cage served as the experimental unit. The authors never stated whether any correction for multiple comparisons was included in the analysis, this needs to be included.

The results are overall improved. In figure 1, the standard error bars need to be included and in the figure legend the authors need to state how many eggs were used to create the average described on each day. Figure legends need to be able to stand alone. Throughout the results section, remove the word significantly and simply state the p value.

The discussion is still overall well-written. I suggest you have a native English speaker read this to double check the grammar as there are still many mistakes.

The intent of the conclusion is overall improved, however much work needs to be done on the accuracy of the grammar. It also is a little confusing, try to present it in a more linear fashion – first discuss how EGF is highest in chicken embryos at d 5 on incubation and then discuss the effects of the animal trial.

Specific comments:

Line 23: this should read: crude extract, respectively.

Line 23: changed sacrificed to harvested

Line 44: The sentence beginning with in recent years should read: In recent years, the poultry industry has ben developing rapidly. However, the more I look at it, the more I realize how vague this sentence is. Some additional supportive material would really strengthen this sentence.

Line 71: animals’

Line 72: remove the word the before novel

Line 76: include the word the before digestive

Line 86: remove the word therefore

Line 94: In the sentence beginning on line 94 the words the and their need to be removed from this sentence.

Line 97: include the word a between in and previously

Line 267: remove the word the from between when and chicken and also from between on and day.

Author Response

This manuscript is a resubmission of an earlier submission. The following is a list of the peer review reports and author responses from that submission.

Round 1

Reviewer 1 Report

The evaluated manuscript is within journal’s scope and may be interesting for readers. It describes an interesting approach in broiler nutrition. Unfortunately, the manuscript could not be completely evaluated because two tables are without data. There are also other shortcomings that requires correction before an acceptance. Research hypothesis was not formulated in the Introduction. Also, the aim of the study was not formulated.

9 - In the Simple Summary and elsewhere “food animals” must be replaced by “animals feed”.

12 - milk-borne growth factor

35 – immunity

38-42. It does not look good when a manuscript about broiler chicken starts with a sentence about pig. It is suggested to move this sentence elsewhere.

56-69 – delete this paragraph because it is not related to the subject of the study.

70 – epidermal (lower-case letter “e”)

78 – Paneth cells

79-82 – delete this sentence

In Material and method section the passive should be used, not the imperative. Thus, language style and grammar must be corrected.

105, 107, and elsewhere – Rotor g force should be given instead of rpm.

128 – “50 birds each”

151 – “…(F/g) was based on…”

158-177 – Type of analyzers (blood analyzer, spectrophotometers and plate reader) used for analyses should be given.

There are no data in Table 2 and Table 3.

208-216 – please replace all “;” by dots.

248-250 – delete

265-267 – The sentence must be rewritten, e.g. “There was also no significant improvement of F/g in our study.” The explanation is not satisfactory. How can the lack of difference in F/g be explained by “difference in addition dose”?

277 – “…serum AKP level significantly increased…

304-306 – This part of the paragraph must be rewritten, e.g. “Enhancing poultry immunity is of great significance for improving poultry health. Poultry immunoglobulins mainly include…There was no significant change in IgG content, but IgA and igM….”

316 – capacity of the immune system…bursa of Fabricius, which is a lymphoid organ…

319 – It is also suggested…

322 – 332 – The Conclusions section should be rewritten, because it is a summary and repetition of results. Only the last two sentences contain conclusion.

 In table 3, it would be better to express uric acid concentration (UA) in micromoles/L, while AKP in U/L. These are the SI units for these parameters.

Reviewer 2 Report

In the manuscript “Effects of dietary supplementation of gEGF on the growth performance and immunity of broilers” the authors had tried to determine the effect of gEGF on growth performance and immunity of broilers. Moreover, the gEGF tried to use as alternative of antibiotics of broiler. Overall, the concept of the manuscript is interesting, but it needs some major clarifications for make it understand to reader. Based upon these major concerns I think this manuscript rewrite for publish in Animals. 

Major comments

  1. The gEGF extraction: Is this your own method? Have you validated it? Then you should refer the paper where you have discussed the method development and validation. If it is an established method, please add the reference.
  2. The explanation of all Figures and Tables are difficult to understand because you explain all figures and tables as percentages, but your figure and table don’t show percentages.

Minor comments

Simple summary

  1. In simple summary, ‘Epidermal growth factor’ is presented 4 times, so use abbreviation.

Abstract

  1. Line 25-29: the result explained by percentages by all of this study presented with different unit.
  2. Line 26: What is UA? Need to full name at first time.

Introduction

  1. Line 39: It says “Introduce”, but “Introduction” should be more suitable.
  2. Line 40-42: please add the reference.
  3. Line 53-55: Need reference

Materials and method

  1. Please show the ethical contents.
  2. How to use the embryo before 6 day? At this time there existed just egg York and white.
  3. Line 105: Use the ‘x g’ unit for centrifuge or explain the machine information. Because the RPM depends on size and speed.
  4. Line 108: please explain how you removed the fat.
  5. Line 118: ‘dried’. How to dry? Evaporation? Air dry? Heat?
  6. Line 120: Please explain the method for measurement of purity.
  7. Line 133: Please explain about cage, as wide x length x height etc.
  8. Line 135: Have any reason for fractional feeding until 2 weeks of age? Please explain.
  9. Line 175: What is cat?

Results

  1. Figure 2, 3, 4. explain about error bar. (std? se? etc).
  2. In Figure 4: It is necessary to classify the superscript that indicate significance by treatment section.
  3. Table 2, 3: Sem should be SEM

Discussion

  1. Line 278-280: That sentence is no need.
  2. Line 291-296: This part is result. Transfer to result section and delete in discussion part.
  3. Line 302: What is ‘to some extent’?
  4. Line 306-307: Please remove ‘there was no significant change, but’
  5. Line 307-310: This sentences are about normal information of IgA and IgM.
  6. Line 324: Remove ‘and’, and insert ‘down regulated and’
  7. Line 329: Explain more clearly. T-SOD is only I liver and CAT is liver and serum.

References

  1. Must change to Animals journal format.

Reviewer 3 Report

The goal of the research presented in this manuscript was to determine at what age chicken embryos have the highest EGF content and then to extract that EGF and include it in the diet of broiler chickens in order to assess differences in growth, feed efficiency, serum metabolites and immune status.

The first sentence in the simple summary is not correct – there is no scientific evidence that shows that using antibiotics in food animals has led to an increase in antibiotic resistance in humans or animals. As a whole, the world is trying to decrease the amount of antibiotics that are used in all aspects (human and animal medicine, production, etc.). Furthermore, in certain countries (such as the EU and US, antibiotic use in poultry is not allowed. Overall, the whole simple summary needs to be re-written to better describe the value of the research that is being conducted. Also, the authors say the current yield of EGF is low – from the simple summary I can’t tell if you are aiming to optimize this procedure as well? The last sentence needs to be re-written as this aim is not well constructed. There are also many grammatical errors – see specific comment below.

More detail needs to be provided in the abstract – how long were the chickens fed for? Relative to when the EGF was given, when were biological samples collected? This information needs to be included to assess the relevance and meaning of the data being presented.

I would suggest re-wording the beginning of the introduction – there is no need to discuss pig disease when you can simply state that the world wide demand for chicken is increasing… there are plenty of citations you can use that support that statement. In the line beginning on page 42 you need to re-phrase this to make it more applicable to the world as a whole – some countries utilize antibiotics in some species to promote growth – not all countries/species do this. In the sentence beginning at line 48 – this sentence is incorrect – that reference does not support what you are stating. That reference simply states that there is controversy as to whether antibiotic use in animals leads to antibiotic resistance in humans. This statement must be rephrased. Along similar lines, make sure that the sentence beginning at line 51 states that most antibiotic resistance in humans comes from antibiotics used by humans. This sentence is misleading. The sentence beginning on line 79 has a weird transition from discussing other research to all of the sudden stating work that was done in this paper and then back to more typical introduction information. I would suggest re-working this. In the sentence beginning at line 83, state what species these experiments were done in. At the end of the introduction, the authors need to state the objectives and hypothesis of the research presented in this manuscript.

In the sentence beginning on line 113 – the day with the highest EGF concentration needs to be stated. In section 2.3.1 the authors need to state that animal care and use guidelines were followed, if this is applicable to your institution. The sentence beginning on line 132 is worded funny, I think this might need to be two separate sentences. How did the authors determine that each chicken was consuming the specified treatment? The paragraph in section 2.3.2 has many grammatical errors and needs to be re-written. Much more information needs to be added to the statistical analysis section – what was the experimental unit?

The results are overall well presented – I just have a few comments. In section 3.2, the authors need to state what the P values are that are associated with the percent increases they are referring to. In table 3, the authors need to define each of the items – a table needs to be able to stand on its own. This is also true for the figure descriptions as well.

The discussion is overall well written. There are a few grammatical errors that are noted below as specific comments.

In the conclusions section I would really like to see some comments on how the results of this research might be applied – i.e. is this a viable option for producers to use in broiler farms?

Specific comments:

Line 11: The sentence beginning on line 11 is worded funny. It should say something more like: As such, the goal of this research was to determine the effects of providing epidermal growth factor to broiler chickens on growth performance, antioxidant capacity, and immune performance of broilers.

Line 26: Define UA

Line 27: Define AKP and CAT

Line 28 define MDA

Line 30: immune organs? I think you mean the immune system or development of organs involved in the immune system.

Line 31: define IgA and IgM

Line 31: demonstrate rather than demonstrated

Line 38: This should read introduction

Line 44: re-word: can also decrease immune function, causing a decrease in disease resistance.

Line 45: rather than saying long time – state how long

Line 107: The sentence beginning on this line has a different tense than the other sentences.

Line 115: the sentence beginning on this line has a different tense than the other sentences.

Line 119: the sentence beginning on this line has a different tense than the other sentences.

Line 157: should read freezer rather than refrigerator

Line 242: add the word the after in

Line 245: add the word the between forms and foregut

Line 247: change to the to on and remove the word of

Line 268: the grammar of the sentence beginning on this line needs to be fixed
